# Electrodegradation of Acid Mixture Dye through the Employment of Cu/Fe Macro-Corrosion Galvanic Cell in Na_2_SO_4_ Synthetic Wastewater

**DOI:** 10.3390/molecules26154580

**Published:** 2021-07-29

**Authors:** Mateusz Kuczyński, Mateusz Łuba, Tomasz Mikołajczyk, Bogusław Pierożyński, Agnieszka Jasiecka-Mikołajczyk, Lech Smoczyński, Piotr Sołowiej, Paweł Wojtacha

**Affiliations:** 1Department of Chemistry, Faculty of Agriculture and Forestry, University of Warmia and Mazury in Olsztyn, Łódzki Square 4, 10-727 Olsztyn, Poland; mateusz.kuczynski@uwm.edu.pl (M.K.); mateusz.luba@uwm.edu.pl (M.Ł.); lechs@uwm.edu.pl (L.S.); 2Department of Pharmacology and Toxicology, Faculty of Veterinary Medicine, University of Warmia and Mazury in Olsztyn, Oczapowskiego 13, 10-719 Olsztyn, Poland; agnieszka.jasiecka@uwm.edu.pl; 3Department of the Electrotechnics, Energetics, Electronics and Automatics, University of Warmia and Mazury in Olsztyn, Oczapowskiego 11, 10-736 Olsztyn, Poland; pit@uwm.edu.pl; 4Department of Industrial and Food Microbiology, Faculty of Food Science, University of Warmia and Mazury in Olsztyn, Cieszyński Square 1, 10-726, Olsztyn, Poland; pawel.wojtacha@uwm.edu.pl

**Keywords:** electrocoagulation and electrooxidation, Fe sacrificial anode, galvanic cell, anodic dissolution, wastewater treatment, acid mixture

## Abstract

Traditional wastewater purification processes are based on a combination of physical, chemical, and biological methods; however, typical electrochemical techniques for removing pollutants require large amounts of electrical energy. In this study, we report on a process of wastewater purification, through continuous anodic dissolution of iron anode for aerated Cu/Fe galvanic cell in synthetic Na_2_SO_4_ wastewater solution. Electrochemical experiments were conducted by means of a laboratory size electrolyzer, where electrocoagulation along with electrooxidation phenomena were examined for wastewater containing Acid Mixture dye. The above was visualized through the employment of electrochemical (cyclic voltammetry and ac impedance spectroscopy techniques) along with instrumental spectroscopy analyses.

## 1. Introduction

Large amounts of different dyes that are present in industrial wastewaters are responsible for serious environmental and health problems [1,2]. Globally, many industries discharge up to 15% of the total dyes used in effluents directly into water bodies. Among various groups of textile dyes, particular attention should be paid to the so-called azo dyes. This specific group of coloring agents is characterized with the presence of one or more azo groups (-N=N-) [3,4,5]. Large concentrations of azo-type dyes in water have a direct impact on the conditions of all organisms living in a biosystem, which, most importantly, are then transferred to human populations. Various studies conducted in recent years have confirmed serious toxicity, as well as mutagenic and carcinogenic effects of some azo dyes, including their metabolites [6,7,8,9,10]. The above, combined with severe environmental pollutions, have led scientists to consider the development of novel, more efficient treatment methods for the dye-based industrial wastewaters.

Dc-powered electrochemical methods have widely been examined and utilized for the remediation of dye-polluted wastewaters. These methods are of significant interest as they combine three parallel processes, namely electrocoagulation, electrooxidation and electroflotation [11,12,13]. Numerous scientific studies have reported total decolorization of wastewaters containing azo dyes by electrochemical means, in the order of 71–99% [14,15,16,17,18,19,20,21,22]. However, despite the high efficiency of such dc-powered systems, they still suffer from excessive power consumption. Nevertheless, this deficiency could be successfully overcome by an electrochemical method based on the galvanic series theory through the application of an appropriately designed galvanic cell (cathode/anode) reactor. This receptacle generates spontaneous current based on sacrificial anode oxidation, along with reduction reaction proceeding on the cathode surface. As compared with formerly described dc-powered methods, the latter design eliminates the presence of expensive dc. infrastructure and electronic control systems, and also radically reduces total consumption of electricity upon wastewater purification (see recently published work by Łuba et al. from this laboratory on Cu/Al alloy galvanic cell-based dye purification in [23]).

During the electrolysis process that occurs inside the Cu/Fe reactor, an iron-made anode undergoes oxidation reaction, which releases soluble ferrous (Fe^2+^) ions that rapidly form iron hydroxide species through the reaction with OH^-^ ions (and highly reactive short-lived ^●^OH radicals that may lead to the cleavage of the dye molecule) generated on the surface of the cathode [24,25,26,27,28,29,30]. During the electrolysis process, the electrolyte’s pH increases. The mechanism of the anodic and cathodic reactions for the Cu/Fe galvanic cell is described by Equations (1)–(3) as follows:

Anodic reaction:Fe _(s)_→ Fe^2+^ +2e^−^(1)

Cathodic reaction:(2)$$O_{2}+2H_{2}O+4e^{-}\to 4{\mathrm{OH}}^{-}$$

Formation of ferrous hydroxide and ferric oxide:(3)$$2{\mathrm{Fe}}^{2+}+4{\mathrm{OH}}^{-}\to 2\mathrm{Fe}{\left(\mathrm{OH}\right)}_{2\left(s\right)}\;\left(+\frac{1}{2}O_{2}\right)\to {\mathrm{Fe}}_{2}O_{3\left(s\right)}+2H_{2}O$$

Hence, Fe dissolution leads initially to the formation of ferrous hydroxide species Equation (3) during the spontaneous reaction of Fe^2+^, cations emerge in the supporting solution during the anodic dissolution process Equation (1) with hydroxyl (OH^−^) anions generated on the cathode surface Equation (2). Then, in the presence of dissolved oxygen, ferrous hydroxide is converted to ferric oxide (Fe_2_O_3_), according to Equation (3).

The aim of this study was to evaluate the effectiveness of a Cu/Fe mild steel galvanic cell in the electrochemical treatment of synthetic wastewater, comprising small, but significant amounts of Acid Mixture (Acid Violet 90 and Acid Red 357 constituents, Figure 1) dye. The examination of the dye’s removal was made by means of the combined effect of the electrocoagulation and surface-electrooxidation processes.

## 2. Results and Discussion

### 2.1. Characterization of Galvanic Cell’s Operation

The electrochemical characterization of the Cu/Fe galvanic reactor was examined in Na_2_SO_4_ supporting electrolyte containing AM (50 mg dm^−3^, pH = 3.05, κ = 15 mS cm^−1^, 10.5 ppm of dissolved oxygen, and T = 20 °C). After an hour of continuous operation, an open-circuit cell’s voltage (ocv) settled at about 0.58 V (Figure 2a).

The electrochemical performance of the Cu/Fe mild steel galvanic cell was examined by measuring the galvanic coupling current (*I*_gc_) intensity. Hence, after 1 h of continuous Cu/Fe cell’s operation, the recorded *I*_gc_ value reached 8.55 mA or 0.29 mA cm^−2^ (Figure 2b). As previously mentioned, this current resulted from the formation of ferrous hydroxide species during the spontaneous reaction of Fe^2+^, cations evolved into the electrolyte during the anodic dissolution process with hydroxyl OH^-^ anions generated on the cathode, according to Equations (1) and (2) above. Moreover, the Fe^2+^ ions could react with electrolyte-dissolved molecular oxygen to form Fe(OH)_3_ species, according to Equation (4) below:(4)$$4{\mathrm{Fe}}^{2+}+10H_{2}O+O_{2}\to 4\mathrm{Fe}{\left(\mathrm{OH}\right)}_{3\left(s\right)}+8H^{+}$$

Then, freshly formed metal hydroxide species attract oppositely charged dye molecules, leading to their neutralization, destabilization, and formation of larger agglomerates that eventually sediment on the bottom of the reaction tank [31,32,33].

### 2.2. Cyclic Voltammetry Electrochemical Behavior of AM Dye

Initially, the electrochemical behavior of Fe mild steel electrode in 0.054 M Na_2_SO_4_ supporting solution was examined by means of the cyclic voltammetry (CV) technique, over the potential range from −1.1 to 0.0 V vs. SCE. Figure 3a shows the third cycle of the CV response for Fe working electrode of the Cu/Fe galvanic cell, carried out in pure and AM-modified 0.054 M Na_2_SO_4_ electrolyte, obtained at 293 K with a sweep-rate of 50 mV s^−1^.

For the cyclic voltammogram recorded in the absence of the AM dye, two irreversible anodic peaks could clearly be observed, namely a low potential and very broad feature, centered at about −0.85 V, and another one positioned at ca. −0.35 V vs. SCE. Thus, the former peak is related to the process of Fe(II) formation, whereas the latter one could be allotted to its further oxidation to form Fe_3_O_4_ or Fe(OH)_3_ surface layers [34,35,36].

Then, having introduced 50 mg dm^−3^ of AM dye into the electrolyte, two well-pronounced oxidation features (centered at ca. −0.56 and −0.42 V) appeared upon cycling towards more positive potentials. It is strongly believed that these anodic patterns correspond to consecutive surface oxidation (commencing with nitrogen-to-nitrogen bond cleavage) of the examined dye’s constituents [37,38,39,40]. Afterwards, as the cycling becomes reversed towards less anodic potentials, a broad cathodic feature is observed over the potential range from −0.8 to −1.0 V (Figure 3a). This feature is most likely related to reduction of the surface-oxidized AM or its intermediates, upon preceding anodic cycle. Most importantly, as the cycling continues, the above-described pattern exhibits a significant shift towards more negative potentials (see Figure 3b). However, as the CV sweep is then limited to the potential range from −0.75 to −1.1 V vs. SCE, formation of the surface-oxidized intermediates is stopped, and the previously observed reduction peak completely disappears from the cyclic voltammogram after only several cycles (see Figure 3c).

### 2.3. Electrochemical ac Impedance Characterization

The ac impedance characterization of iron electrode in the presence of 50 mg dm^−3^ of AM dye in the supporting 0.054 M Na_2_SO_4_ solution is shown in Table 1 and Figure 4a,b. Hence, in the presence of the AM dye, the Nyquist impedance plots exhibited two “depressed” and partial semicircles over the examined potential range from −950 to −380 mV vs. SCE. Thus, for the most negative potentials (i.e., −950 and −900 mV), the high-frequency semicircle was associated with oxidative formation of Fe^2+^ cations, whereas the low-frequency arc observed in the Nyquist impedance plots corresponded to the surface formation of iron II hydroxide layer (see Figure 4a along with its Bode plot inset). Thus, for the first, −0.95 V oxidation peak (Figure 3a), the recorded charge transfer resistance (*R*_ct_) parameter for iron oxidation came to 67.4 Ω cm^2^, whereas the recorded adsorption, i.e., Fe(OH)_2_ charge transfer resistance (*R*_Ads_) value, reached 1980 Ω cm^2^. Moreover, the corresponding interfacial capacitance (*C*_dl_) and pseudocapacitance (*C*_Ads_) parameters reached 536 and 5181 µF cm^−2^, respectively (see equivalent circuit used to fit the recorded data in Figure 5). It should be noted that these results are in line with those recently reported on a similar experimental azo dye purification system in [41].

The potential range from −630 to −380 mV (Table 1) is characteristic of further iron oxidation to form Fe_3_O_4_ or Fe(OH)_3_ species along with consecutive surface electrooxidation of the dye’s components (see Figure 3a). Hence, the high frequency arcs, observed in the Nyquist plots of Figure 4b, correspond to the former process, whereas intermediate/low frequency semicircles refer to the latter, i.e., dye electrodegradation steps. Thus, for the peak potential points (−570 and −410 mV), the recorded *R*_ct_ reached 32.8 and 12.8 Ω cm^2^, whereas the *C*_dl_ parameter reached 672 and 510 µF cm^−2^, respectively. Interestingly, the recorded charge-transfer resistance (*R*_ox_) values, corresponding to the successive, oxidative electrodegradation of the dye components were radically diminished (6.64 and 2.06 Ω cm^2^, correspondingly). The above comes in line with major augmentation of the recorded pseudocapacitance, *C*_p_ parameter (see Table 1 for details). 

Alternatively, for the pure Na_2_SO_4_ solution at positive potentials to −800 mV, only one somewhat depressed semicircle was noticed in relation to a single-step charge transfer reaction associated with further electrode surface oxidation process, resulting in the formation of Fe(III) species [36] (see examples of this behavior in Appendix A in Appendix A).

### 2.4. UV-Vis Spectrophotometry Analysis

The process of Fe surface electrooxidation and -degradation of the examined AM azo dye upon operation of the Cu/Fe galvanic macro-corrosion cell was primarily monitored by utilizing the UV-Vis spectrophotometry technique, where the process’s optimization was carried out by recording absorption spectra over a 230–800 nm wavelength range (see Figure 6a).

The effect of an initial dye decomposition stage recorded at a wavelength of 500 nm is illustrated in Figure 6b. Here, the recorded, gradually diminishing absorbance, is directly linked to the concentration reduction of the pigment. During the stage of decolorization, the azo (-N=N-) bond, which is a chromogenic entity, becomes an acceptor for electrons released during the anodic oxidation process. Thus, the molecule breaks down into aromatic/phenol and aromatic amine derivatives, which are structurally similar to 3-amino-2-hydroxy-5-nitorbenzenesulfonic acid or 2-chloro-4-nitrobenzamine [37,38,39]. With respect to the aromatic ring degradation stage (λ = 280 nm), over the initial 900 s of the electrolysis process, the recorded changes in the absorbance intensity at the wavelength of 280 nm (Figure 6c) indicate that total concentration of the aromatic type substances/molecules containing aromatic ring in their structures has increased. These results are in good agreement with the above-depicted AM degradation pathway. As it comes to the last step of the AM mineralization process (monitored at λ = 240 nm, Figure 6d), a vast number of relatively simple (e.g., maleic, formic, and acetic acids; aldehydes; ammonia; nitrogen oxides; and finally, carbon dioxide molecule) or more complex chemical compounds (e.g., pentanoic acid) could be produced. Finally, the total quantitative removal of the AM dye through the electrooxidation and electrocoagulation processes yielded 21 and 37%, after 900 and 3600 s of continuous electrolysis, respectively (see Figure 6e).

Furthermore, Table 2 presents the total quantitative removal of various azo dyes, obtained through other novel degradation methods described in the literature [17,18,20,22,40,41,42]. On the one hand, the estimated total quantitative removal of azo dyes for yeast- and algae-based were inferior as compared with the method presented in this study. On the other hand, the literature-based electrochemical methods powered with external power supply turned out to be significantly superior in dye degradation process as compared with both biological and galvanic-driven methods. Nevertheless, they have their own disadvantages, for example, high energy consumption (high operational current densities), as well as use of scarce and rare materials and generation of harmful by-products during the electrolysis process. However, employment of Cu/Fe macro-corrosion galvanic cell excludes the use of an external power supply, with potentially harmful chemicals, along with a relatively low-cost technical infrastructure.

### 2.5. Chromatography

The successive removal of the AM dye from synthetic wastewater solution was also evidenced by ultra-performance liquid chromatography, coupled with double quadrupole tandem mass spectrometry (UPLC-MS/MS) system analysis. The above showed that, just after 900 s of continuous electrolysis, the total removal of the dye’s constituents reached 60 and 71% for AV90 and AR357, respectively. Then, after one hour of the test, the removal of the AV90 and AR357 dye components came to 86 and 91%, respectively (see Table 3 and Figure 7 for details). As compared with the UV-Vis technique, there were significant differences in the recorded data, which resulted from selectivity variation between these two analytical methods.

Appendix A in the Appendix A presents the UPLC-MS/MS chromatogram with the scan showing a number of high molecular mass (MM), unidentifiable electrodegradation products. As the electrolysis process causes the formation of small amounts, but high MM compounds, it could be concluded that after initial breaking of the -N=N- azo bond, some of the reactive fragments of an initial dye could react with each other, thus, forming such species.

## 3. Materials and Methods

Construction of Galvanic Cell Reactor, Chemicals, and Solutions Experimental Methodologies.

In this study, the wastewater electrocoagulation/electrooxidation unit was composed of ca. 300 mL glass-made electrochemical reactor (see Figure 9 in [23]) and electrodes, which were arranged in a single Fe sacrificial anode placed between two Cu cathodes. A cylindrically shaped Fe mild steel was used as a sacrificial anode with an effective surface area of 29.3 cm^2^ (diameter Φ = 3.5 cm, thickness: d = 1 cm) (DIN C15 (1.0401), ArcelorMittal, Dąbrowa Górnicza, Poland) and cathodes were made of cylindrically shaped copper with a total effective electrode area of 84 cm^2^ (two plates 4 × 5 × 0.1 cm). An anode-to-cathode interelectrode gap was about 15 mm.

All procedures for the preparation of Cu and Fe electrodes, as well as electrochemical cell and working solutions’ preparation, employed electrochemical instrumentation and methodologies (cyclic voltammetry and ac impedance spectroscopy), and the UV-Vis spectroscopy analyses were as those recently reported in [23,43] from this laboratory. The electrolyte pH, dissolved oxygen concentration, and conductivity evaluations were performed with pHenomenal^®^ pH 1100 L, pHenomenal^®^ OX 4100 L, and pHenomenal^®^ CO 3100 L meters from VWR, correspondingly. The chromatographic quantitation of AR357 and AV90 levels was achieved in reversed-phase liquid chromatography ACQUITY UPLC (ultra-performance liquid chromatography) system with I-Class Plus coupled with a Xevo TQ-XS Triple Quadrupole Mass Spectrometry (Waters, Milford, CT, USA). Chromatographic separation of AR357 and AV90 dye components was performed by means of ACQUITY UPLC BEH C18 column (1.7 μm, 2.1 × 50 mm, Waters) maintained at 50 °C. The mobile phase consisted of phase A (0.005% ammonium acetate for AR357 and AV90) and phase B (acetonitrile for AR357 and AV90) in the gradient elution. Each sample analysis was carried out for 5 min, at a flow rate of 400 µL min^−1^. An injection volume was 1 µL and temperature of the autosampler was maintained at 15 °C. Detection was performed with a double quadrupole tandem mass spectrometer in the negative ion mode for both AR357 and AV90 pigments. The setting parameters of the detector are illustrated in Table 4 below. Equipment was set up in a multiple reaction monitoring (MRM) mode and transitions of 447.5 and 442.5 m/z were used for determination of the AR357 and AV90 parts, respectively. Samples collected from electrolysis (100 µL of AM) were dissolved in water to achieve final volume of 500 µL and transferred into total recovery vials (Waters). Finally, an aliquot was injected for UPLC-MS/MS analysis.

Acid Mixture (Acid Violet 90 and Acid Red 357, p.a.) dye was supplied by Boruta-Zachem S.A. (Bydgoszcz, Poland). In order to provide the required high effectiveness of surface electrooxidation processes, conductivity of the working solutions was settled at about 15 mS, the working solution was under additional aeration to obtain ca. 10.5 ppm of dissolved oxygen content, whereas their pH was adjusted to the set value of 3.05. 

## 4. Conclusions

The electrochemical method, based on the operation of a macro-corrosion, Cu/Fe galvanic cell, for wastewater purification from AM dye pollutant, referred to in this work, gave very encouraging results. Surface electrooxidation (degradation) of the dye components seems to be a key and necessary step to initiate the process of the dye removal from synthetically made wastewater.

The surface electrooxidation reactions were strongly supported by the recorded ac impedance spectroscopy and cyclic voltammetry results. In addition, the UV-Vis and UPLC-MS/MS characterizations implied a very complex and unpredictable mechanism of AM electrodecomposition, most likely leading to the formation of numerous, though extremely hard to identify, products. 

In summary, in this study, the proposed wastewater treatment method, based on the operation of commonly available macro-corrosion galvanic cell, may be considered to be a suitable replacement for the generally used, multi-step wastewater treatment methods. However, further work is necessary in order to adapt this technology to real industrial conditions.

## 5. Patents

Pierożyński, B. and Smoczyński, L., Electrocoagulator for wastewater treatment, Patent of the Republic of Poland, PAT.227874, granted on 5 September 2017.

## Figures and Tables

**Figure 1 molecules-26-04580-f001:**
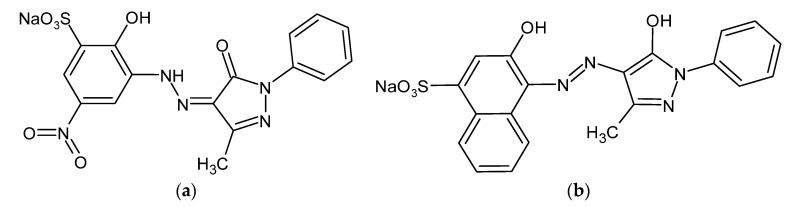
Chemical structure of the acid mixture (AM): (**a**) Acid Violet 90 (AV90); (**b**) Acid Red 357 (AR357), azo dyes molecules.

**Figure 2 molecules-26-04580-f002:**
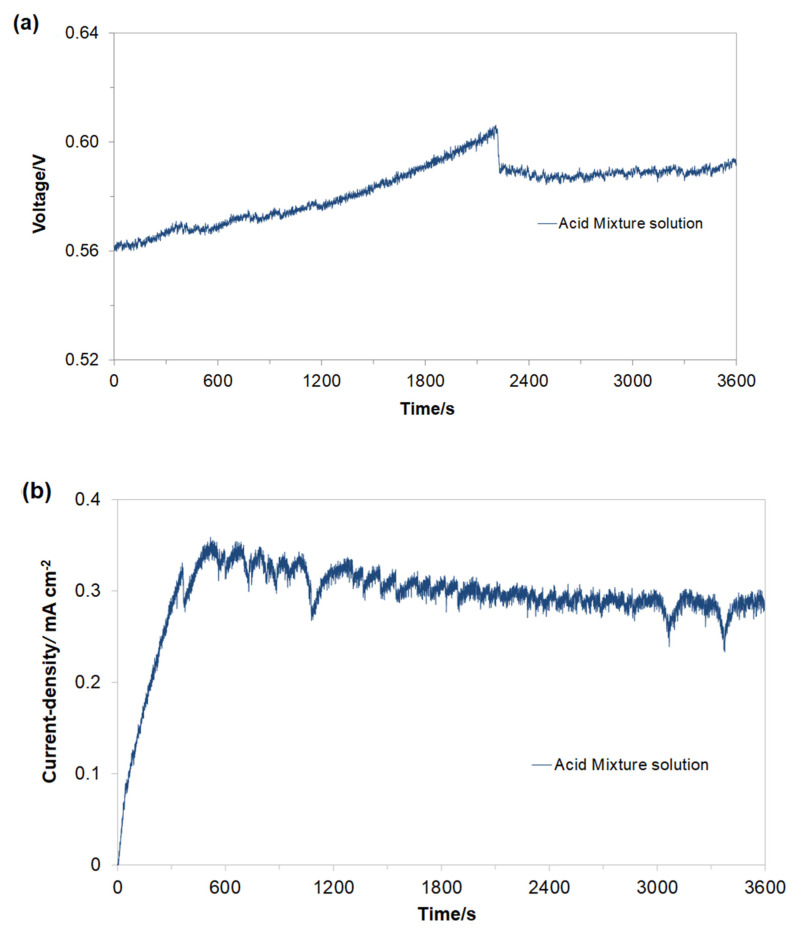
Recorded values of (**a**) open-circuit voltage and (**b**) galvanic couple current-density for macro-corrosion Cu/Fe galvanic cell in function of exposure time, derived for freshly prepared and aerated Na_2_SO_4_-based solution, in the presence of AM dye, at a concentration of 50 mg dm^−3^.

**Figure 3 molecules-26-04580-f003:**
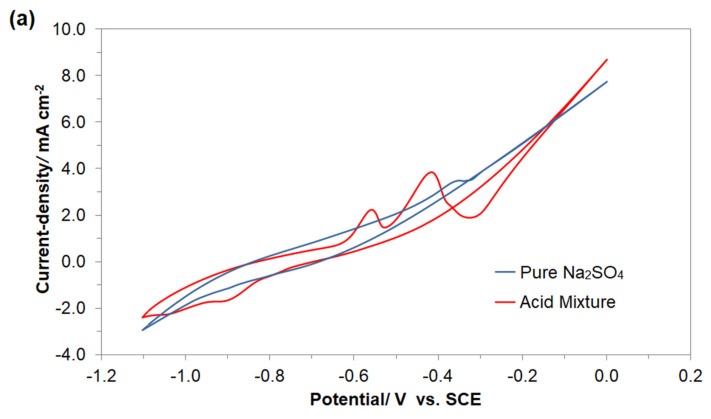
(**a**) Cyclic voltammetry behaviour of the Cu/Fe galvanic cell in the absence and presence of AM dye (at 50 mg dm^−3^), recorded in 0.054 M Na_2_SO_4_ supporting electrolyte, at a sweep-rate of 50 mV s^−1^ and 293 K (third cycle); (**b**) shift of the peak potential value for the recorded cathodic reduction process upon continuous cycling; (**c**) disappearance of cathodic reduction peak upon cycling over the potential range narrowed to −1.1–(−0.75) V.

**Figure 4 molecules-26-04580-f004:**
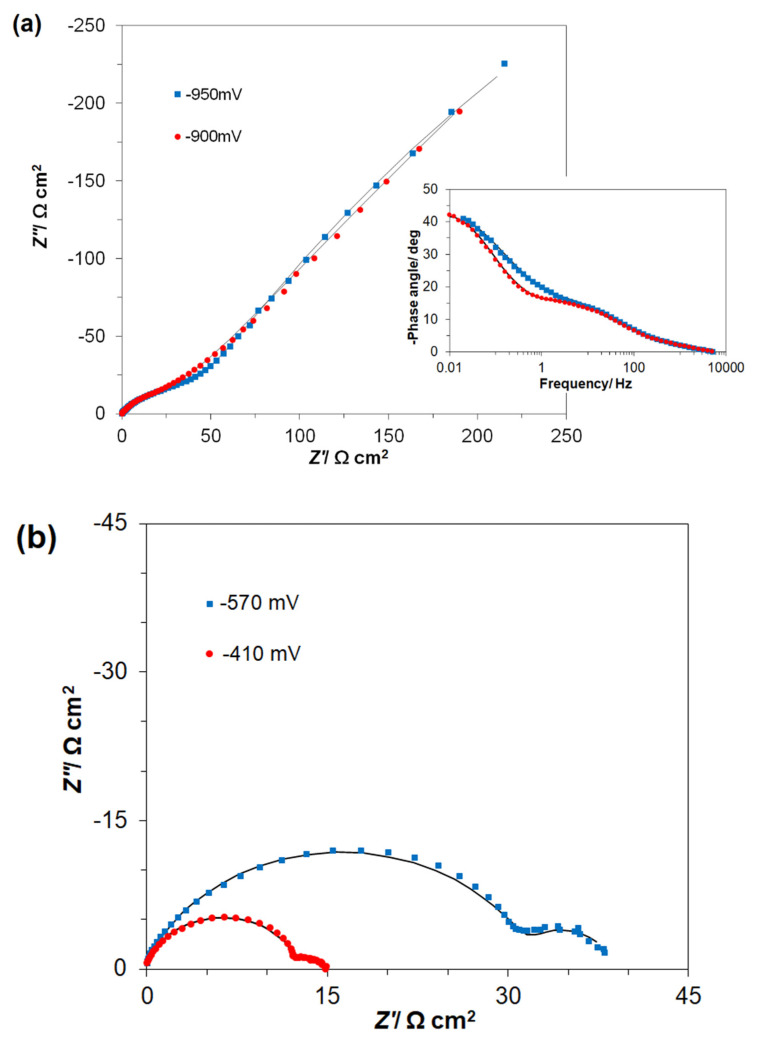
The recorded complex plane Nyquist impedance plots for low carbon steel electrode in contact with AM-modified (50 mg dm^−3^) Na_2_SO_4_ solution, derived for the stated potential values vs. SCE, where (**a**) corresponds to the surface formation of iron II hydroxide layer and (**b**) corresponds to iron oxidation to form Fe_3_O_4_ or Fe(OH)_3_ species along with consecutive surface electrooxidation of the dye. The solid lines correspond to the representation of the data according to the equivalent circuit model presented in Figure 5.

**Figure 5 molecules-26-04580-f005:**
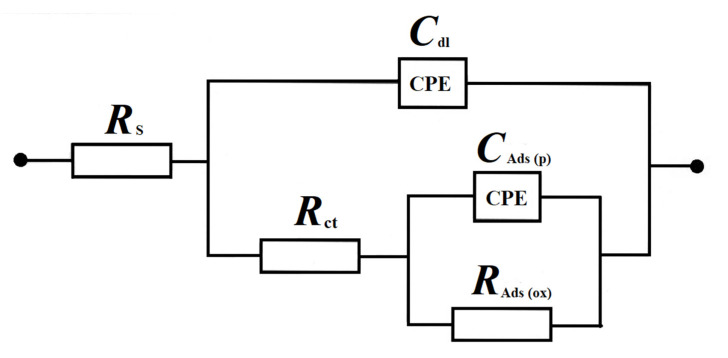
Equivalent circuit model used for fitting the obtained ac impedance spectroscopy data, where *R*_s_ is solution resistance; *C*_dl_ is double-layer capacitance; *R*_ct_ is charge-transfer resistance parameter for electrooxidation of Fe anode surface; *R*_Ads_ and *C*_Ads_ are adsorption resistance and capacitance parameters for Fe(OH)_2_ film, respectively; and *R*_ox_ and *C*_p_ are azo dyes electrooxidation resistance and pseudocapacitance parameters, respectively, connected with oxidative degradation of *AM* dye molecule. The circuits include constant phase elements (CPEs) to account for distributed capacitance.

**Figure 6 molecules-26-04580-f006:**
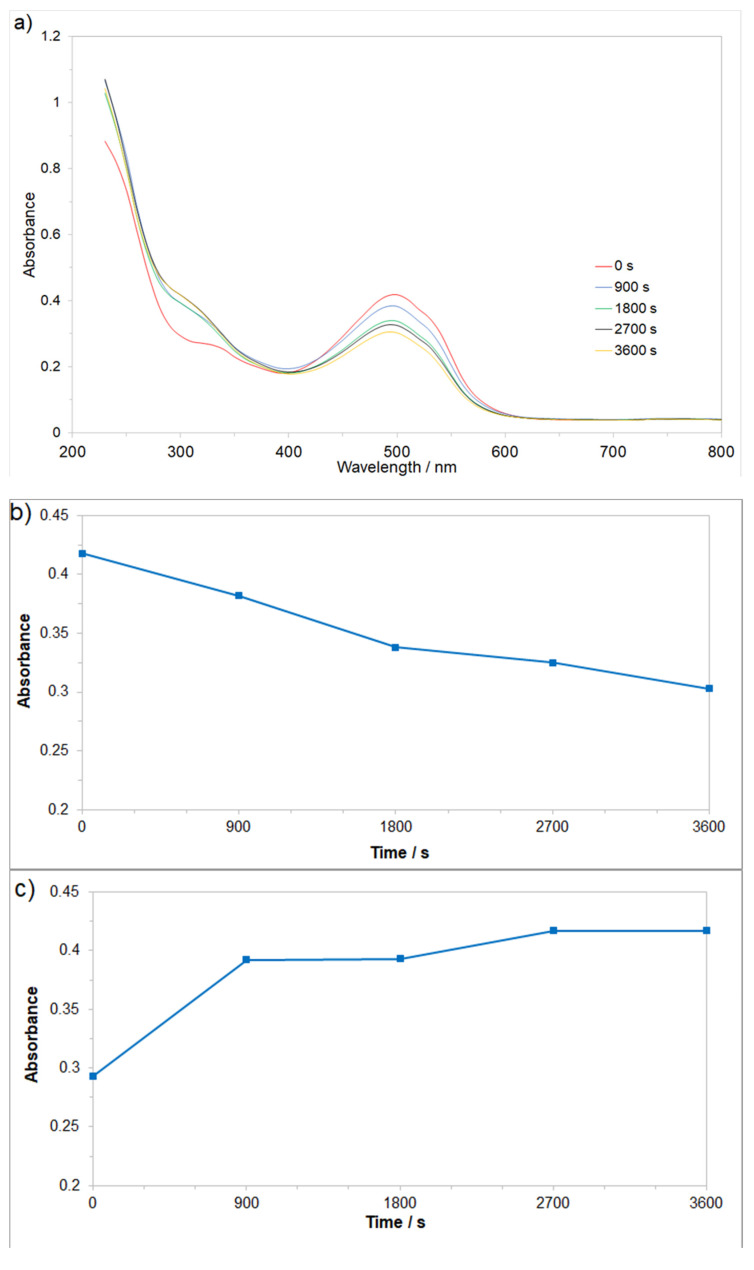
UV-Vis absorption spectra changes in time for AM dye’s electrodegradation, carried-out by means of Cu/Fe galvanic cell, where: (**a**) Representation of the three segments of AM dye’s electrooxidation process on the surface of Fe anode in Na_2_SO_4_ solution (for an initial dye concentration of 50 mg dm^−3^); (**b**) evolution of decolorization process, recorded for a wavelength of 500 nm; (**c**) progress of aromatic ring degradation, recorded for a wavelength of 280 nm; (**d**) evolution of mineralization process, recorded for a wavelength of 240 nm; (**e**) quantitative dye assay after sedimentation process, recorded for a wavelength of 500 nm.

**Figure 7 molecules-26-04580-f007:**
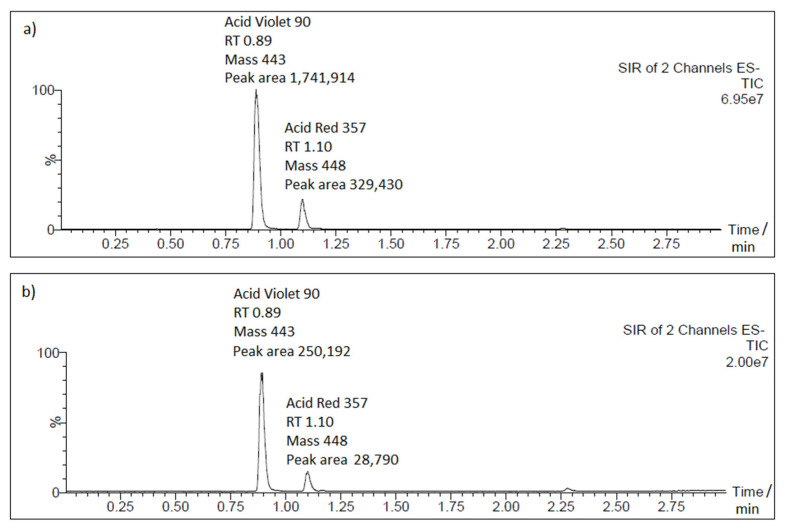
Representative UPLC-MS/MS chromatograms for AM: (**a**) before; (**b**) after 60 min of continuous electrolysis.

**Table 1 molecules-26-04580-t001:** Parameters for the processes of Fe electrode oxidation and azo dye electrodegradation (at a total concentration of AM at 50 mg dm^−3^) on the surface of the Fe mild steel anode in contact with 0.054 M Na_2_SO_4_, achieved by fitting equivalent circuit models presented in Figure 5 to the experimentally obtained impedance data (dimensionless φ parameter, which determines the constant phase angle in the complex-plane plot (0 ≤ φ ≤ 1) of the CPEs circuit, varied between 0.79 and 0.86, and 0.50 and 0.96 for φ_1_ and φ_2_, respectively).

*E*/mV	*R*_ct_/Ω cm^2^	*C*_dl_/µF cm^−2^ s ^φ1-1^	*R*_Ads_/Ω cm^2^	*C*_Ads_/µF cm^−2^ s ^φ2-1^
*Acid Mixture*				
−950	67.4 ± 10.5	536 ± 174	1980 ± 498	5181 ± 818
−900	40.2 ± 3.3	1828 ± 116	5857 ± 1574	2991 ± 410
			***R*** **_ox_** **/Ω cm^2^**	***C*** **_p_** **/µF cm^−2^** **s ^φ2-1^**
−630	52.6 ± 1.1	604 ± 32	28.4 ± 2.5	35,372 ± 2581
−570	32.8 ± 0.3	672 ± 21	6.6 ± 0.4	152,788 ± 12,425
−450	15.1 ± 0.2	464 ± 14	2.8 ± 0.1	157,761 ± 11,691
−410	12.8 ± 0.1	510 ± 25	2.1 ± 0.2	243,297 ± 24,982
−380	12.1 ± 0.1	603 ± 38	1.4 ± 0.2	400,853 ± 72,947

**Table 2 molecules-26-04580-t002:** Efficiency comparison and features of various novel azo dyes removal methods.

Method of Azo Dyes Removal	Type of Process	Total Removal %	Exposure Time	Use of External Power Supply	Current Density/mA cm^−2^	Use of Toxic Chemicals	Ref.
Cu/Fe macro-corrosion galvanic cell	Physico-chemical	37%	1 h	No	0.29	No	This work
Novel yeast consortium	Biological	6–16%	1 h	No	-	Heavy metals	[40]
Electrochemical degradation with BDD-anode	Physico-chemical	*ca*. 40%	1 h	Yes	30.00	No	[22]
Tank with *Chlorella vulgaris* microalgae	Biological	*ca*. 18%	2 days	No	-	No	[41]
Electrochemical treatment with Aluminum anodes	Physico-chemical	87%	1 h	Yes	-	Yes (Cl_2_ formation)	[17]
Electrochemical treatment with Sb-doped SnO_2_ ceramic electrodes	Physico-chemical	ca. 50%	1 h	Yes	15.00	No	[18]
Electrochemical oxidation using RuO_2_-IrO_2_ coated titanium electrode as an anode followed by biodecolorization using *Pseudomonas stutzeri* MN1 and *Acinetobacter baumannii* MN3	Combination of physico-chemical and biological methods	98%	10 min + 24 h (Electrochemical + biological techniques)	Yes	20.00	Yes (Cl_2_ formation)	[20]

**Table 3 molecules-26-04580-t003:** Variation of the detected chromatographic peak areas for AV90 and AR357 substrate components by time of electrolysis.

Electrolysis Time/min	Peak Area	
	AV90	AR357
0	1,741,914 ± 37,287	329,430 ± 6872
15	696,022 ± 44,056	96,908 ± 3107
30	460,865 ± 25,491	66,638 ± 4787
45	349,217 ± 13,973	40,856 ± 3795
60	250,192 ± 15,038	28,790 ± 340

**Table 4 molecules-26-04580-t004:** Tandem mass spectrometry (MS/MS) parameters.

MS/MS Parameters	
Precursor ion (*m*/*z*)	447.5 (AR357) 442.5 *(*AV90)
Product ions (*m*/*z*)	-
Desolvation gas	nitrogen
Desolvation gas temperature (°C)	350
Desolvation gas flow (L/h)	300
Cone gas flow (L/h)	150
Collision gas	argon
Source’s temperature (°C)	120
Electrospray mode	negative
Cone voltage (V)	20
Capillary voltage (kV)	3.0
Retention time (min)	0.89 (AV90) 1.10 (AR357)

## Data Availability

The data presented in this study are available on request from the corresponding author.

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
