# Peer review of "Electrodegradation of Acid Mixture Dye through the Employment of Cu/Fe Macro-Corrosion Galvanic Cell in Na2SO4 Synthetic Wastewater"

_molecules, 2021, doi:10.3390/molecules26154580_

Round 1
Reviewer 1 Report
This work is well done and presented. I recommend its publication after the following revision.
In my opinion, the degradation sequence of the the dyes should be proposed and discussed, based on the detected reaction intermediates.
Reviewer 2 Report
This work would be required major revision for further consideration.
- This work demonstrates an electrochemical decomposition of two azo dyes. The application of the present technique is limited, while the title 'Electrodegradation of acid mixture...' is quite broad. So, I'd suggest addition of extensive applications using the process.
- Compare the efficiency of the present electrochemical method with the previously reported methods (similar procesure using electrodegradation). Without comparision, the advantages of the present method is not clear.
- How to support the possible degradation mechanism of the dyes. If possible add detail data for characterization of the electrochemical degradation.
- The meaning of figure 7 showing representative UPLC-MS/MS is not so clear. Please add more data to demonstrate time-dependant changes (degradation) of the dyes.
Reviewer 3 Report
The electrodegradation of acid mixture dyes (Acid Violet 90 and Acid Red 357) by using Cu/Fe macro-corrosion galvanic cell in Na2SO4 syntheric wastewater is studied in this manuscript. This manuscript is unsuitable published in Molecules under the comments addressed in the followings.
- In page 9, authors inferred that the decolourization is caused by the anodic oxidation of the azo (-N=N-) bond, which is unreasonable under the reasons given in the followings:
1-1. Based on the results of cyclic voltammetry (CV), the peak potential of anodic oxidation of Fe (-0.85 V) is lower than that of anodic oxidation of dyes (-0.56 V). Hence the main anodic reaction of Fe electrode should be the anodic oxidation of Fe, but not the anodic oxidation of dyes.
1-2. The results of CV can not be used to explain the results of declourization of dyes in terms of Fe/Cu galvanic cell due to the charges within CV came from the power supply. However, the currents (i.e. charges) are came from the spontaneous reactions given in Eqs. 1 and 2, that is came from the dissolution of Fe, but not from the electro-oxidation of dyes.
1-3. Hence the decolourization of dyes should be the coagulation and flocculation of dyes by the presence of Fe(OH)2.
- The method for measuring the dissolved oxygen (i.e. 10.5 ppm) should be addressed in the manuscript.
- In Fig. 2:
3-1. The phenomena of the sudden decrease of the voltage of the galvanic cell at about 2100 ~ 2200 s in Fig. 2a should be described and explained.
3-2. The reasons for the sharp increase of the current density in the initial state, and the slow decrease of the current density for the time greater than 600 s should be discussed.
- The semi-circles in the high and intermediate frequencies in Fig. 4b are attributed to the oxidation of iron and dye, respectively. The reasons should be discussed.
Round 2
Reviewer 1 Report
The revised paper is ready for publication.
Reviewer 2 Report
I found that the revised paper is improved and contains useful infomration, although some contents in this article are still not clear. The method will require extensive optimization process for real-world applications in the future. I'd recommend accept of this article.
Reviewer 3 Report
The authors' answer to the comments of mine still cannot convince me.